# Process Characterization of Polyvinyl Acetate Emulsions Applying Inline Photon Density Wave Spectroscopy at High Solid Contents

**DOI:** 10.3390/polym13040669

**Published:** 2021-02-23

**Authors:** Stephanie Schlappa, Lee Josephine Brenker, Lena Bressel, Roland Hass, Marvin Münzberg

**Affiliations:** 1Department of Physical Chemistry, innoFSPEC, University of Potsdam, Am Muehlenberg 3, 14476 Potsdam, Germany; josie.brenker@googlemail.com (L.J.B.); bressel@uni-potsdam.de (L.B.); rh@pdw-analytics.de (R.H.); marvin.muenzberg@uni-potsdam.de (M.M.); 2PDW Analytics GmbH, Geiselbergstraße 4, 14476 Potsdam, Germany

**Keywords:** photon density wave spectroscopy, multiple light scattering, emulsion polymerization, process analytical technology, polyvinyl acetate

## Abstract

The high solids semicontinuous emulsion polymerization of polyvinyl acetate using poly (vinyl alcohol-co-vinyl acetate) as protective colloid is investigated by optical spectroscopy. The suitability of Photon Density Wave (PDW) spectroscopy as inline Process Analytical Technology (PAT) for emulsion polymerization processes at high solid contents (>40% (*w*/*w*)) is studied and evaluated. Inline data on absorption and scattering in the dispersion is obtained in real-time. The radical polymerization of vinyl acetate to polyvinyl acetate using ascorbic acid and sodium persulfate as redox initiator system and poly (vinyl alcohol-co-vinyl acetate) as protective colloid is investigated. Starved–feed radical emulsion polymerization yielded particle sizes in the nanometer size regime. PDW spectroscopy is used to monitor the progress of polymerization by studying the absorption and scattering properties during the synthesis of dispersions with increasing monomer amount and correspondingly decreasing feed rate of protective colloid. Results are compared to particle sizes determined with offline dynamic light scattering (DLS) and static light scattering (SLS) during the synthesis.

## 1. Introduction

Emulsion polymerization systems are of high importance to the chemical industry. Products like adhesives, sealants, and coatings are produced from emulsion polymerization processes at large scales. Polyvinyl acetate (PVAc) is one of the most common polymers. One possibility of producing PVAc at industrial scale is protective colloid stabilized emulsion polymerization. In such systems, the amphiphilic protective colloid forms a colloidal system that acts as reaction center in which the polymerization process takes place [1]. Oligomers formed in the water phase from monomer molecules and the water-soluble initiators diffuse into the protective colloid micelles and react there to polymer chains with further monomer molecules diffusing into the micelle. Emulsion polymerization provides numerous advantages like being a greener process [2], possessing low viscosity throughout the process, providing advanced heat transfer, and a narrower particle size distribution (PSD) than regular suspension polymerization processes [3,4,5]. For the industry the control of the PSD is of paramount interest to obtain specifically designed products as the PSD determines the characteristics and properties of such dispersions [6].

To control the particle size there are several options in emulsion polymerization. The amount of protective colloid influences the PSD and application properties of the produced dispersion. The more protective colloid in the system, the more reaction centers are formed, and a larger number of particles with a smaller size can be obtained [1,7,8]. The type of protective colloid also plays an important role. Various types of protective colloids are currently under investigation. Polymeric or plant based emulsifying agents can be used to increase dispersion stability [7]. A common protective colloid used for the synthesis of PVAc is polyvinyl alcohol poly (vinyl alcohol-co-vinyl acetate) (PVA) due to its ability to form colloidal aggregates with different physico-chemical properties. Various studies on the blockiness, size, and interfacial tension behavior of formed PVA colloidal aggregates have been carried out [8,9,10]. These studies have shown, that the colloidal aggregates formed, strongly depend on the hydrophobic–hydrophobic interactions between the vinyl acetate sequences in the PVA and this strongly correlates with the degree of hydrolysis. As the above-mentioned properties of the PVA colloidal aggregates influence the particle size during synthesis of PVAc the hydrolysis degree therefore plays an important role for particle size control.

An additional possibility to control the PSD of the final product is via starved–feed synthesis. A starved–feed system guarantees nearly complete conversion of monomer to polymer [1,4,5], the possibility to obtain high solid contents in the final product and control of the particle size by addition of monomer to the reactor. Low feed rates of monomer in starved–feed conditions control the rate of particle growth. Using an initial charge without any monomer content forces particle growth to happen under starved-monomer conditions and particle growth to be fully controlled by the monomer feed rate. Stopping particle growth can instantaneously be achieved by stopping the addition of monomer to the emulsion system and desired particle sizes for specified applications can be targeted [11,12]. Further possibilities for size control include feed rates or types of initiators, the monomer, or the protective colloid [13,14,15,16,17,18]. As the relationship between particle size and all these parameters is complex, a simple approach to predict the exact evolution of the PSD from all these parameters is not possible. Here, monitoring the progress of the reaction inline would be desirable.

Due to the high turbidity already at low polymer solid contents, commonly used inline Process Analytical Technologies (PAT) to monitor reaction progress or even particle size (distributions), like turbidity probes or optical inline microscopy are limited in their application and are less suitable to investigate polymerization processes [19,20,21]. To monitor the reaction processes or particle growth in turbid emulsion polymerization processes, Photon Density Wave (PDW) spectroscopy has shown to be a reliable method under certain conditions [22,23] by measuring the optical scattering properties. PDW spectroscopy is a new inline PAT that determines the absorption coefficient *µ*_a_ as well as the reduced scattering coefficient *µ*_s_’ of the dispersion in real time without dilution or sampling. Here, physical and chemical characteristics of the analyzed sample can be calculated from the optical coefficients [24,25,26,27,28,29]. In a recent study [23] the evolution of the particle size during the synthesis of PVAc was monitored by PDW spectroscopy and increasing particle sizes were observed as expected. However, sudden steps in the obtained particle size during the synthesis revealed, that the particle size analysis for PVAc systems during synthesis might be more complex than anticipated.

In this study, the suitability of PDW spectroscopy as inline PAT to monitor the reaction progress from the optical coefficients in the highly turbid emulsion polymerization of PVAc is explored and PDW spectroscopy is introduced as an applicable method for real-time measurements in emulsion polymerization even without particle size analysis. The possibility to monitor the synthesis inline via the optical coefficients could already lead to enormous advantages like better reproducibility, consistent product properties and quality, as well as reduction of waste and cost.

## 2. Materials and Methods

Chemical reagents were used as follows: Monomer vinyl acetate (VAc, ≥99%, Sigma-Aldrich, Darmstadt, Germany) was purged with N_2_ (Nippon Gases, Duesseldorf, Germany) for 30 min prior synthesis. Redox initiator pair ascorbic acid (AA, 99%, Acros Organics, Geel, Belgium) and sodium persulfate (NaPS, ≥99%, Carl Roth, Karlsruhe, Germany) as well as the catalyst ammonium iron (III) sulfate hexahydrate (FAS, 99+ %, Acros Organics, Geel, Belgium) were used as purchased. Mowiol^®^ 4-88 (poly(vinyl alcohol-co-vinyl acetate), PVA, approx. 31.000 g mol^−1^, viscosity of a 4% solution at 23 °C is 4 mPAS, 86.7–88.7 mol% hydrolysis degree, Sigma-Aldrich, Darmstadt, Germany), was used as purchased. All solutions were prepared with analytical grade Milli-Q^®^ water from an in-house Milli-Q^®^ water dispenser (Milli-Q^®^, Integral 5, Merck Millipore, Darmstadt, Germany) and purged with N_2_ for 30 min prior synthesis.

### 2.1. Synthesis of Polyvinyl Acetate Dispersions

The synthesis described here is based on the industrial process to produce vinyl ester polymers described in [30]. Various research has been done on the polymerization process, e.g., regarding the type or amount of initiators, the onset temperature or the types of copolymers [13,14,15,16,17,18]. All syntheses described here were performed in an automated lab reactor (OptiMax 1001, Mettler Toledo, Gießen, Germany) at 1 L reaction scale. Reactions were executed at 60 °C under nitrogen atmosphere applying reactor temperature control, stirring control, dosing control, overhead reflux condenser, and N_2_ purging. Dosing of feeds was achieved using four computer controlled automated dosing units (SP-50, Mettler Toledo, Gießen, Germany) for vinyl acetate (two dosing units) and approx. 16.7% (*w*/*w*) PVA in water (two dosing units) as well as two automated syringe pumps (PHD ULTRA, Harvard Apparatus, Holliston, MA, USA) with one syringe for the redox initiator AA and one for NaPS. Redox initiator aqueous solutions of 3.5% (*w*/*w*) AA and 4.5% (*w*/*w*) NaPS were used for all syntheses. A scheme of the experimental set-up in shown in the Appendix A. Five experiments were performed with a varied amount of total monomer fed to the system at a constant feed rate and decreased feed rates of PVA. This results in the same duration of dosing PVA and monomer while keeping the total PVA amount (*V*_PVA_ = 190 mL) fed to the system constant. The protective colloid PVA Mowiol^®^ 4-88 was chosen to obtain particle sizes in the nanometer size regime. The chosen PVA has further advantages as it offers good solubility at the reaction temperature of 60 °C, is relatively dormant to hydrogen bonding and stays in solution upon cooling [31].

To start a synthesis 300 g Milli-Q^®^ water were filled into the reactor and 0.33 g PVA (= 1 g L^−1^) were dissolved under stirring and heating up to 60 °C. The reactor was closed, and the initial charge was purged with N_2_ for 30 min. After complete dissolution of PVA in the initial charge, approx. 0.018 g FAS in 2 mL of water were added in order to catalyze the redox system. After adding the FAS catalyst, the syringe pump was started feeding AA and NaPS into the reactor at 0.33 mL min^−1^ each. After approx. five minutes of initiator feeding, monomer and PVA feeding were started according to Table 1. PVA feeding was stopped simultaneously to the feeding of monomer. Redox initiators were dosed for 10–30 min after VAc and PVA feeding ended in order to achieve complete conversion.

An additional synthesis PVAc_1wr analogous to synthesis PVAc_1 but with a reduced initial amount of 151.2 g of water, an increased amount of 5.52 g PVA (=36.3 g L^−1^), and an amount of 0.009 g of FAS in 2 mL of water, keeping the concentration of FAS in the initial charge constant, was carried out. Here, the same feeding rates as for synthesis VAc_1 for monomer, PVA (15.7% (*w*/*w*)) were used. A total amount of 350 mL of monomer, 190 mL of PVA solution, and 31.6 mL of each initiator was targeted. During this synthesis, the feeding of monomer and PVA had to be stopped before finishing the targeted amount. Table 2 displays targeted and actually dosed amounts. This synthesis was carried out to evaluate how a much higher amount of protective colloid and a total increase in solid content would affect inline measurements of PDW spectroscopy and whether process control via PDW spectroscopy would still be possible.

All syntheses were monitored in line with PDW spectroscopy regarding the trend of the optical coefficients. Offline-samples were taken for solid content and particle size analysis. Additionally, the density and refractive index of the particles in the final dispersion were analyzed. All syntheses described above were carried out a single time.

### 2.2. Analysis of Polymer Emulsions

For offline analysis, 1 mL samples were taken manually out of the reactor at approx. 40 mm depth using a single-use 1 mL syringe with a cannula length of 13.5 cm, diluted approx. 1:10 with Milli-Q^®^ water to quench the reaction and stored in sample containers. For process analysis until *t*_R_ = 5 min samples were taken every minute, until *t*_R_ = 30 min samples were taken every 5 min, until *t*_R_ = 120 min samples were taken every 10 min and for *t*_R_ > 120 min samples were taken every 20 min.

Gravimetric solid content analysis was done with approx. 500 µL sample stored in weighted Eppendorf tubes and dried at 72 °C for at least 24 h until mass consistency. After drying, the total weight of the solid residue was taken and handled as solid content of the dispersion at sampling time *t*. Samples were weighted without purification, solid content refers to all possible solid residues after drying.

Offline sample analysis was performed after the synthesis was completed. Dynamic light scattering (DLS) analysis (Zetasizer Ultra, Malvern Panalytical, Worcestershire, United Kingdom) was done at a measurement angle of 173° using single-use 4 mL polystyrene cuvettes at 25 °C. Samples were diluted approx. 400-fold to visual transparency. For static light scattering (SLS) measurements (LS13320, Beckman Coulter, Brea, CA, USA) samples were pre-diluted approx. 125-fold with Milli-Q^®^ water and pipetted into the sample chamber. Using Polarization Intensity Differential Scattering (PIDS) as measuring method, the sample was pipetted into the reaction chamber until a PIDS signal of 40% was achieved. For high quality analysis, low obscuration values beneath 2% were retained.

For data analysis of PDW spectroscopy measurements, the densities and the refractive indices of the particles and the dispersant are needed. To determine the density of the particles a concentration series of 30% (*w*/*w*), 20% (*w*/*w*), 10% (*w*/*w*), 1% (*w*/*w*), and 0.1% (*w*/*w*) of each polymer dispersion was measured at 25 °C with a densitometer (DM45 Delta Rage, Mettler Toledo, Gießen, Germany). The data were analyzed for the density of the particles (i.e., pure polymer, 100% (*w*/*w*)) using the following equation [22,32]:(1)$$\rho_{\mathrm{Disp}}=\frac{\rho_{D}\rho_{P}}{w\left[\rho_{D}-\rho_{P}\right]+\rho_{P}}\;↔\;\frac{1}{\rho_{\mathrm{Disp}}}=w\left[\frac{1}{\rho_{P}}-\frac{1}{\rho_{D}}\right]+\frac{1}{\rho_{D}}$$

Here, *ρ*_Disp_, *ρ*_D_, and *ρ*_P_ are the densities of the dispersion, the dispersant, and the polymer and *w* is the solid content of the dispersion. For the density of zero polymer content (0% (*w*/*w*)) the density of pure water was considered, determined with the same densitometer. The density of the polymer was obtained from the slope of a linear fit according to the right equation. The error was obtained via error propagation.

The same concentration series was used for refractive index measurements at 20 °C at seven different wavelengths between 403 and 938 nm with a multi-wavelength refractometer (DRS-λ, Schmidt + Haensch, Berlin, Germany). The measured refractive indices are extrapolated to the refractive index of the particle using the Newton equation [28] and afterwards inter- or extrapolated to the wavelengths of PDW spectroscopy by a Cauchy formula [22,32].

Solid content, particle size, density, and refractive index analysis were carried out with three repeating experiments each. Values and errors shown in the results section reflect mean values and standard deviations from these repeating experiments.

### 2.3. Photon Density Wave Spectroscopy

Photon Density Wave (PDW) spectroscopy is a technique for the characterization of the optical properties of multiple light scattering materials, such as highly turbid polymer dispersions [27]. Using optical fibers, intensity modulated laser light is guided into the sample. Due to multiple scattering and absorption in the material, a PDW is created, which alters in amplitude and phase while passing the turbid material. The changes to amplitude and phase of the PDW are characterized as function of the modulation frequency and distance *r* between emission and detection fiber and can be related to the absorption coefficient *µ*_a_ and the reduced scattering coefficient *µ*_s_’ of the material.

Whereas the absorption coefficient *µ*_a_ can be attributed to the concentration of absorbing species in the sample, the reduced scattering coefficient *µ*_s_’ can in principle be attributed to the amount and size of the scattering material and is dependent on the physical properties like density and refractive index of the particles and dispersant. By using *µ*_s_’ physical changes within the dispersion can be monitored. Previous research showed, that monitoring the scattering properties of various samples like milk, milk-fat, zeolites, or bioplastic producing bacteria leads to deeper insights into the processes and deeper understanding of underlying mechanisms like depletion flocculation, phase transitions or zeolite production [24,25,26]. For polymer syntheses monitoring *µ*_s_’ is an important tool, as it mirrors polymerization phases and hence offers possibilities for process control. Additionally, applying Mie theory and theories for dependent light scattering [28] the size of the dispersed particles can in principle be derived from the reduced scattering coefficient in inline measurements [33,34]. For polydisperse systems a multi-wavelength approach is necessary [28]. However, due to the complexity of the PVAc system, containing high amounts of highly water soluble PVA as protective colloid, which forms a non-negligible protective layer on the particle surface rendering a simple PVAc-particle-in-a-medium-of-water model moot, particle size analysis based on PDW spectroscopy is beyond the scope of the present work.

Inline PDW spectroscopy was applied using a specially designed inline probe with eight optical fibers and a probe diameter of 25 mm. The probe was inserted into the reactor via an inlet in the reactor lid with the tip of the fibers’ positions at approx. 40 mm beneath the initial charge–air-interface, to obtain stable PDW data from the beginning of the experiment. The PDW spectrometer used here is self-made. Commercial versions are available from PDW Analytics GmbH, Potsdam, Germany. Detailed technical description of the set-up can be found elsewhere [32,33]. Measurements for process characterization were executed at 637 nm, 690 nm, and 751 nm with modulation frequencies up to 1210 MHz and fiber distances between 2.4 and 19 mm. As syntheses were performed only once. PDW data shown here are therefore also obtained from a single measurement. Errors shown for *µ*_a_ and *µ*_s_’ are obtained from a non-linear fit process during the analysis of intensity and phase [35].

## 3. Results and Discussion

### 3.1. Dispersion Analysis

Stable, highly turbid liquid dispersions were obtained in all cases. Drying a droplet of the dispersion at ambient air conditions leads to a transparent, hard, non-sticky film. Stored in airtight plastic containers, the dispersions are storable without any phase separation for at least six months. Dilution of the samples with Milli-Q^®^ water was possible without phase separation or visible agglomeration of the particles. All produced dispersions exceeded 40% (*w*/*w*) solid content. Electron microscopy (cf. Appendix A) revealed that spherical, well separated particles were produced in all syntheses.

Figure 1 shows the gravimetrically derived solid content during the five syntheses PVAc_1 to PVAc_2. For all syntheses the solid content increases as long as monomer is added and then levels off. It reflects an evolution of the solid content typically achieved with free radical polymerization processes [36]. Following chain-growth mechanism, mostly monomer reacts with the active radical chain creating high instant conversion of monomer to polymer of 90 to 98% (data not shown). Propagation of polymer chains is the dominant process until no more monomer is added to the system and propagation stops. Calculated solid content assuming the direct conversion of monomer to polymer is show in Figure 1. All experimental data match the theoretically calculated solid content fortifying the assumption of a starved–feed synthesis. Only for synthesis VAc_2 the solid content shows some fluctuations at high solid contents, as sampling and drying of these samples is challenging. The synthesis VAc_2 was the only one which led to a little amount of coagulate at the interface between air and dispersion in the final dispersion. Due to the high amount of total monomer dosed, visible agglomerates were formed. It is assumed, that the synthesis VAc_2 with 700 mL monomer forms visible agglomerates, because the amount of protective colloid fed during the synthesis is not sufficient to stabilize the formed particles. The highest feedable amount of monomer to create a completely stable dispersion is therefore somewhere between synthesis VAc_1.8 with 630 mL and synthesis VAc_2 with 700 mL of dosed monomer under the reaction conditions presented here.

Figure 2 shows the mean of the gravimetrically determined final solid content of all five dispersions from three repeating measurements. The final solid content increases with increasing monomer dosing as expected. Calculations for 100% conversion of monomer to polymer lead to an expected maximal solid content in the final dispersion *w*_calc, PVAc_ of 36.7% (*w*/*w*) to 51.1% for VAc_1 to VAc_2. As the high molecular protective colloid anchors to the particle surface, it does not evaporate during the drying process. The gravimetrically determined solid content therefore consists of both polymer and protective colloid. The solid content calculated from 100% conversion of monomer to polymer including additionally the total amount of PVA in the dispersion *w*_calc, PVAc+PVA_ leads to values between 40.75% and 54.08% for syntheses VAc_1 to VAc_2. These calculated values match the experimental solid contents well. In case of VAc_2 the experimental solid content is slightly smaller than the calculated value. This can be anticipated due to particle agglomeration as discussed above.

The extrapolated polymer density of the particles of (1.20 ± 0.01) g cm^−3^ (Table 3) as mean of all synthesized dispersions shows very good agreement between all batches and is in agreement with data from literature, where a density of 1.17 to 1.2 g cm^−3^ for PVAc homopolymers, depending on the degree of polymerization, is found [37]. Extrapolated refractive index values of the particles for a wavelength range from 400 nm to 1000 nm are shown in Appendix A. Values between 1.48 and 1.49 at *λ* = 637 nm were extrapolated from dilution measurements for the pure polymer particle. For the synthesis VAc_2 with aggregated particles no determination of the density nor the refractive index was possible. Therefore, density and refractive index values obtained for VAc_1.8 were used for PDW spectroscopy analysis of VAc_2.

### 3.2. Polymerization Monitoring Using PDW Spectroscopy

Figure 3 exemplarily shows the inline monitoring of synthesis VAc_1.5. The system is first heated to a reaction temperature of 60 °C. The overshoot of a few degrees over 60 °C is due to insufficient PID correction of the automated reactor. To overcome long inhibition periods, the redox initiators were fed to the system five minutes before starting monomer dosing (not displayed here), to provide radicals for the instant nucleation and propagation of polymer chains. The polymerization is started by adding monomer VAc and protective colloid PVA to the reaction system (*t*_R_ = 0 h). The start of the polymerization process can be observed in the temperature profile in Figure 3 by a characteristic release of reaction heat at *t*_R_ = 0 h. A sudden increase in temperature of approx. 4 K is observed. During synthesis the reaction temperature stays nearly constant at 60 °C until approx. *t*_R_ = 2 h where the addition of VAc and PVA is finished and a slight drop in the reaction temperature (less pronounced than the increase at *t*_R_ = 0 h) occurs due to the final conversion of monomer to polymer and the termination of release of reaction heat. The temperature drop happens slightly after the finish of monomer and PVA feed indicating that small amounts of monomer are still converted to polymer. This small time lag however still is in accordance with the assumption of a starved–feed synthesis. After 2.7 h, the system is cooled down to 20 °C over 30 min.

Monitoring the synthesis in line with PDW spectroscopy allows to observe changes in the optical properties of the sample, like the absorption coefficient *µ*_a_ and the reduced scattering coefficient *µ*_s_’. With dosing monomer to the system, *µ*_a_ at λ = 637 nm drops from 0.004 mm^−1^ to a local minimum of around 0.0008 mm^−1^ at 1.5 h. *µ*_a_ then increases slightly again and levels off, as soon as monomer dosing is stopped. With start of cooling at 2.7 h, again a slight decrease in *µ*_a_ is observed with a subsequent levelling off as soon as 20 °C is reached.

The reduced scattering coefficient *µ*_s_’ in Figure 3, obtained at λ = 637 nm, increases rapidly as soon as monomer and protective colloid are added. After about 20 min, *µ*_s_’ increases less pronounced and a maximum value of approx. 25 mm^−1^ is reached at *t*_R_ = 1.5 h, simultaneously to the absorption minimum in *µ*_a_. After the maximum, a decrease to approx. 24 mm^−1^ is observed and *µ*_s_’ levels off in correspondence to stopping monomer and protective colloid dosing. Until start of cooling at *t*_R_ = 2.7 h, *µ*_s_’ stays constant. Cooling down the system to 20 °C causes an increase in *µ*_s_’ to significantly higher values of around 27 mm^−1^.

Three different phases are represented in the trend of the reduced scattering coefficient *µ*_s_’. First, *µ*_s_’ rises rapidly until *t*_R_ = 0.2 h with a very steep slope, which might be due to fast particle formation and growth. After the initial steep increase *µ*_s_’ continues to increase, with a less steep slope. During this second stage, until start of cooling, particles should still be growing as monomer is continuously added. During the cooling process, *µ*_s_’ increases again and finally *µ*_s_’ levels off, as the sample reaches room temperature and the cooling process is completed. The increase of *µ*_s_’ during cooling might be due to temperature dependent changes of physical properties of the particles like density and refractive index. In this case, e.g., an increase in particle density, leading also to an increase in refractive index, would result in a decrease of particle size. Depending on the amplitude of each effect, the result can either be a decrease or increase of *µ*_s_’.

These stages can be observed for all syntheses carried out here. The inline measured reduced scattering coefficient for all five syntheses is shown in Figure 4 (cooling stages are omitted for better visibility). Each polymerization starts at *t*_R_ = 0 h with the start of dosing monomer and protective colloid into the reactor. The initial steep increase of *µ*_s_’ appears to be identical for all syntheses, and hence, seems not to be affected by the decreased feeding rate of PVA to the system.

Except for VAc_1, all syntheses exhibit a maximum in *µ*_s_’ during the feeding of monomer and protective colloid. The maximum occurs at nearly the same time, a slight shift to the left with increasing monomer amount is visible. In emulsions with a lower amount of protective colloid, less reaction centers for particle formation and growth are formed. Therefore, fewer, but bigger particles are growing in the dispersion which should lead to an increase of light scattering at these still relatively small particle sizes in this early stage of synthesis. Although feeding of monomer and PVA is not yet finished, *µ*_s_’ then decreases again. This might be due to the distinctive dependency of *µ*_s_’ on particle size, refractive index, and volume fraction. Described by Mie theory, *µ*_s_’ exhibits oscillations as function of particle size for given chemical and optical properties. These oscillations change their form and position in dependency of the measurement wavelength. Accordingly, as shown in Figure 5 for VAc_1.5 and VAc_1.8, shifts of the maximum in *µ*_s_’ are observed when applying measurements at different wavelengths. Hence, the presence, position, and height of these maxima contains the key to particle sizing in these systems during synthesis.

Due to the complex dependency of *µ*_s_’ on particle size or particle size distribution and due a non-linear dependency on solid content, known as dependent scattering, particle sizing during the reaction at high solid contents and elevated temperatures is challenging. Here, changes in density and refractive index of the particles and medium might occur. Apart from the complexity of the PVAc-PVA-water system as mentioned before, determining all physical properties in the addressed temperature range, and including models for dependent light scattering might be therefore necessary to obtain correct particle sizes.

Stopping the feed of PVA and monomer corresponds well to reaching a plateau phase in *µ*_s_’ in both syntheses (Figure 5). At the end of the syntheses *µ*_s_’ exhibits different values, depending on wavelength and total amount of monomer. For synthesis VAc_1.5 the final *µ*_s_’ increases with decreasing wavelength. For synthesis VAc_1.8 measurements at 690 nm and 751 nm end at nearly the same *µ*_s_’ value of approx. 23 mm^−1^, a higher value is obtained at 637 nm. Despite the different solid content in the final dispersions *µ*_s_’ exhibits nearly the same value of 26 mm^−1^ for synthesis VAc_1.5, VAc_1.8, and VAc_2 (cf. Figure 4) at 637 nm. The difference in the end values arises due to the underlying particle size distribution. To understand the exact form of *µ*_s_’ in dependency of wavelength during the polymerization a light scattering model for the reaction needs to be set up. The evaluation of particle size, assuming pure polymer or additionally PVA as particle and water as surrounding medium seems currently not to be sufficient as data analysis model. Here, a more sophisticated model is required to describe the experimentally observed reduced scattering coefficients.

### 3.3. Comparison to Offline Size Analysis

Even though particle sizing still is challenging due to an insufficient optical model of the PVAc system, both the reduced scattering coefficient and the absorption coefficient contain process-related information. As soon as monomer and PVA addition is stopped, both coefficients level off, indicating the end of the reaction. In case of *µ*_a_ this might be obvious as no more absorbing chemicals are consumed or absorbing products formed. However, in case of *µ*_s_’ a levelling off does not only reflect the end of dosing but also the constancy of the size (distribution) of scattering material. In addition to monitoring changes of the reaction temperature to obtain information about the synthesis, e.g., a short increase at the beginning and a short decrease at the end of the feeding, the optical coefficients are therefore a powerful tool to monitor the reaction progress inline. This is especially helpful when temperature changes are not visible or significant, as is the case for synthesis VAc_1.8 (cf. Figure 5 right). In another case, where at the end of feeding the monomer is completely converted to polymer, a drop in temperature would suggest the end of the reaction. However, if aggregation between particles occurred, *µ*_s_’ would still show changing values (cf. Section 3.4 and Figure 7).

To undermine the assumption of comparing particle growth to the reduced scattering coefficient, offline analysis of the particle size has been carried out with DLS and SLS. Results are shown exemplarily for VAc_1.5 in Figure 6. In the beginning the inline trend in *µ*_s_’ and both offline determined particle diameters start to increase rapidly. The sharp slope in the first PDW measurements corresponds directly to the particle growth up to *t*_R_ = 0.2 h. First measurements in the early minutes of the reaction showed particle sizes of approx. 50 nm, agreeing to the theory, that colloidal aggregates of PVA formed to stabilize the particles are smaller than 50 nm in size [8,9]. The second, less rapid growth period of particles is also reflected in a more modest growth phase of *µ*_s_’ until the maximum value is reached. A slight decrease in *µ*_s_’ is evident, although still growth of the particles is apparent in DLS and SLS measurements. Hence, the maximum might derive from the assumed dependency of *µ*_s_’ on particle size as described above. After *t*_R_ = 2 h the monomer and protective colloid dosing is stopped and particle growth is inhibited. During cooling of the dispersion starting at *t*_R_ = 2.7 h *µ*_s_’ increases even though DLS and SLS values stay constant. It has to be pointed out that DLS and SLS data are always obtained after cooling to room temperature, whereas PDW spectroscopy measurements are performed at reaction temperature. The final particle diameter after *t*_R_ = 3.7 h differs by approx. 140 nm between both offline measurements. While DLS provides a final *d*_P_ of (619 ± 12) nm, SLS results in a value of (480 ± 1) nm (error indicates the standard deviation of three repeating measurements). The results obtained by DLS represent the hydrodynamic radius, whereas SLS determines the geometrical diameter of the particles.

In principle, determining mean particle sizes directly in line with PDW spectroscopy is feasible [24,33,36]. However, due to the complexity of the system and the temperature dependent physical properties particle sizes might not yet be accessible. Further research needs to be executed for the correct description of wavelength dependent light scattering in this complex, size distributed, and concentrated particle system.

### 3.4. Inline Process Control Using PDW Spectroscopy

To show that PDW spectroscopy can in principle also be applied as technique for process control, a synthesis with a decreased volume of the initial charge but high PVA content is shown in Figure 7. Total amounts fed to the system are shown in Table 2. Here, only 150 mL of water were used as initial charge to increase the amount of PVA in the initial charge drastically. With continuously adding more monomer to the system the polymerization proceeds and particles continue to grow in the dispersion. Apart from the initial peak at *t*_R_ = 0 h, during synthesis the reaction temperature stays nearly constant at 60 °C until approx. *t_R_* = 1.1 h where two sudden peaks of approx. 8 K to higher temperatures are observed shortly after each other. This rise in temperature is caused by gelation of the sample as gelation hinders the heat transfer in the sample.

Monitoring the synthesis in line with PDW spectroscopy reveals a sudden drop at the same time in *µ*_a_ as well as *µ*_s_’. *µ*_s_’ then increases even steeper than before parallel to the two temperature peaks. Due to the observed temperature peaks and drops in the optical coefficients, the addition of monomer and PVA was stopped while continuing the initiator feeds to prevent further gelation in the sample. This resulted in the reaction temperature to decrease to 60 °C again. Initiator dosing was continued to decrease both viscosity and to achieve complete conversion of residual monomer. Shortly after returning to 60 °C and discontinuing the monomer and PVA feeds, the absorption coefficient levels off. At the same time the reduced scattering coefficient does not level off, but increases further, however, with a lees steep slope. After *t_R_* = 2.1 h the reaction was cooled down to 25 °C within 30 min after dosing of the initiators was finished. *µ*_s_’ only levels off shortly after room temperature is reached. This indicates that the process is not yet finished when then monomer and then initiator feedings are stopped even though particle sizes measured offline did not change anymore after monomer feed is stopped. Further conversion of monomer to polymer in the monomer swollen particles might be the reason. The indirect dependency of *µ*_s_’ on temperature due to its dependency on refractive index, density, and particle size might be an explanation for the sudden drop in *µ*_s_’ during the peak in temperature in Figure 7, in contrast to the syntheses shown before (cf. Figure 3, Figure 4 and Figure 5) where no such sudden drop in either optical coefficient is observed. The steeper slope after the drop might arise due to multiple reasons, such as changes in the sample properties, faster particle growth or particle agglomeration. Due to the gelation of the sample the synthesis was aborted and PVA and VAc feed were stopped earlier than planned. Because of the higher amount of water, no gelation occurred in the other syntheses shown above.

This synthesis shows that the optical coefficients can directly be used to control the process, i.e., in this case, stop the monomer feed. Additionally, a second possibility for process control could have been the continuing increase of *µ*_s_’ during the cooling process. Here, the reaction was cooled down according to the reaction plan 20 min after the dosing of initiator feeds was finished. As *µ*_s_’ was still rising at that time this change could have been used as an indicator to continue the reaction at 60 °C until *µ*_s_’ levels off. This could prevent incomplete conversion of monomer to polymer in the final product. The invers is also possible in batch processes. Here, typically the reaction temperature is held constant for a certain time which is obtained from experience to obtain complete conversion, these tempering periods are often longer than required. The monitoring of *µ*_s_’ could help to decrease these unnecessary long reaction times by cooling down directly after *µ*_s_’ reaches a plateau.

## 4. Conclusions and Further Research

A series of starved–feed emulsion polymerizations of vinyl acetate dispersions with high solid contents of up to 54% (*w*/*w*) and high turbidity with increasing amount of monomer and decreasing feeding rate of PVA was successfully carried out. The reaction progress as well as particle size monitored with different inline and offline techniques is presented. Shown trends in solid content were in good agreement with expected reaction progress known from emulsion polymerization. To verify the high conversion rates in future experiments, gas chromatography (GC) could be applied to monitor the conversion of monomer to polymer.

PDW spectroscopy was applied to monitor the reaction progress inline throughout the whole polymerization, via the absorption and scattering properties. Absorption and scattering properties were successfully monitored and compared to polymerization stages evolving during emulsion polymerization. Three stages could be identified, agreeing well to emulsion polymerization theory. A first stage of initial particle formation and fast particle growth is reflected by a steep increase in the reduced scattering coefficient for all syntheses. In the second phase of synthesis, during particle growth, *µ*_s_’ rises slower, depending on the feeding rate of protective colloid. In some syntheses, maxima are expressed, which might reflect the formation of different particle size distributions of polymer particles in the dispersions. Levelling off of *µ*_s_’ during the third phase occurred as soon as feeding of chemicals was stopped and polymerization terminated. The subsequent cooling of the synthesis caused an increase in *µ*_s_’, probably due to temperature dependent physical properties, like refractive index and density of polymer particles and dispersant.

Comparison of inline *µ*_s_’ with particle sizes obtained offline from DLS and SLS confirmed these three phases of the reaction, i.e., initial steep increase of particle size, further slower particle growth and finally levelling-off of the particle size. To obtain particle sizes or even distributions from PDW spectroscopy during the synthesis, as has been shown in previous studies [27,34], further studying of the system is necessary to set up an improved scattering model for the complex PVAc–PVA–water system. This includes temperature dependent physico-chemical properties like density and refractive index of all components, determination of monomer to polymer conversion, e.g., with GC as well as a revision of the particle-water model.

In an additional synthesis with increased protective colloid amount, it was shown, that from combination of temperature data as well as inline PDW spectroscopy measurements of the optical coefficients process control is possible. Here, gelation of the sample was observed and the reaction stopped by stopping the monomer addition. Further gelation was hindered by adding dispersant, in this case low concentrated initiator solutions.

Both examples clearly show the capability of monitoring polymerization progress during synthesis of vinyl acetate via PDW spectroscopy. This allows for a better understanding of the synthesis as well as control of the reaction which will improve the quality of the obtained product and the prevention of waste batches.

## Figures and Tables

**Figure 1 polymers-13-00669-f001:**
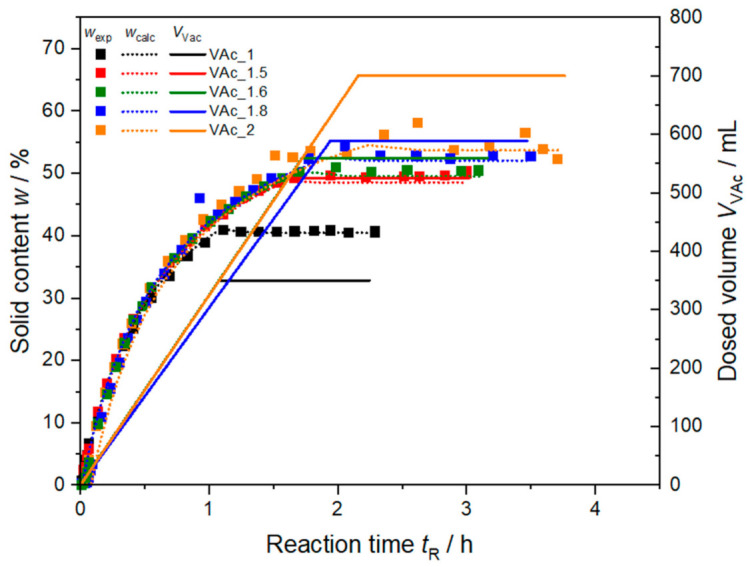
Gravimetrically determined and calculated solid content *w*_calc_ from PVAc and PVA combined and dosed volume of monomer *V*_VAc_ during polymerization for syntheses VAc_1, VAc_1.5, VAc_1.6, VAc_1.8, and VAc_2.

**Figure 2 polymers-13-00669-f002:**
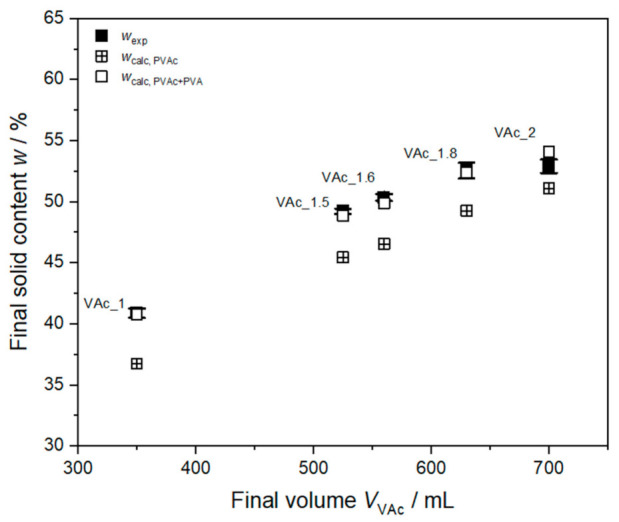
Final experimental solid content of all syntheses (full squares) in dependency of the total dosed volume of monomer *V*_VAc_ compared to the calculated solid content for 100% monomer to polymer conversion *w*_calc, PVAc_ (crossed squares) and calculated solid content for 100% conversion of monomer to polymer and attachment of the total amount of PVA to the particle *w*_calc, PVAc+PVA_ (open squares).

**Figure 3 polymers-13-00669-f003:**
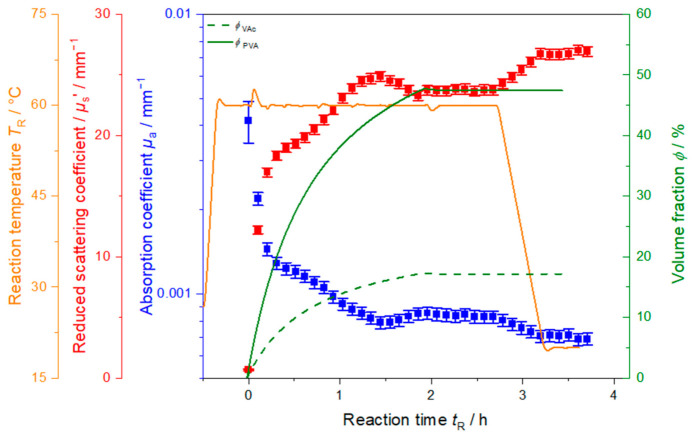
Reduced scattering coefficient *µ*_s_’ (red) and absorption coefficient *µ*_a_ (blue) at 637 nm, volume fraction of monomer *ϕ*_VAc_ and protective colloid *ϕ*_PVA_ (green solid and dashed lines, respectively), and reaction temperature *T*_R_ (orange) as function of time during polymerization of VAc_1.5.

**Figure 4 polymers-13-00669-f004:**
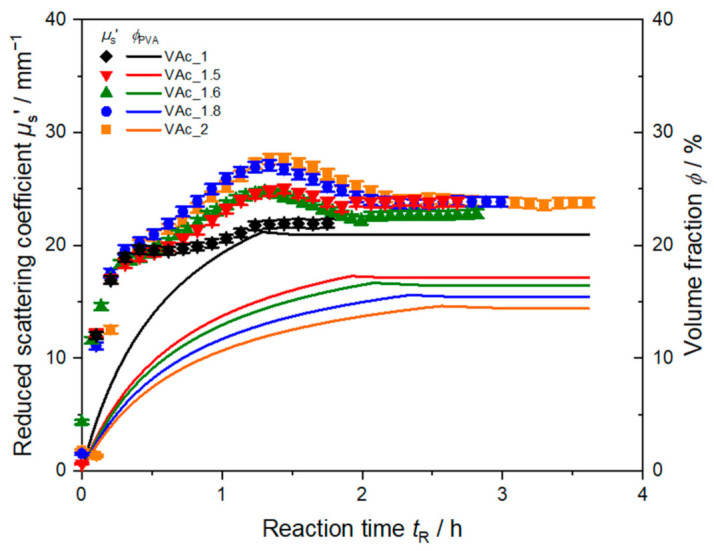
Comparison of inline determined reduced scattering coefficient *µ*_s_’ at 637 nm for synthesis PVAc_1 to PVAc_2 with different feeding rates of PVA and hence volume fraction *ϕ*_PVA_ during synthesis. Data shown without cooling phase for better visualization.

**Figure 5 polymers-13-00669-f005:**
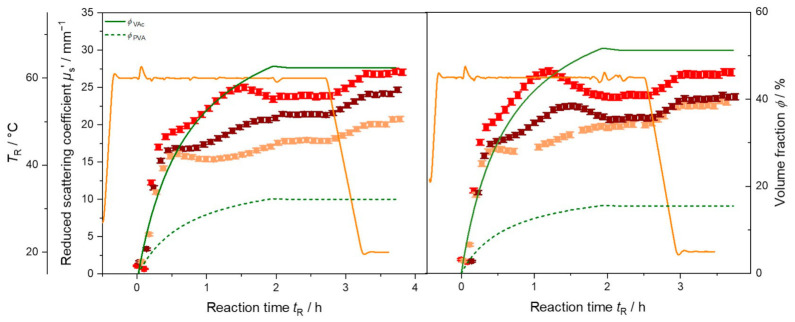
Reduced scattering coefficient *µ*_s_’ (symbols) at *λ* = 637 nm (red), *λ* = 690 nm (brown) and *λ* = 751 nm (orange), volume fraction of dosed monomer *ϕ*_VAc_ and protective colloid *ϕ*_PVA_ (solid green and dashed lines, respectively), and reaction temperature *T*_R_ (orange line) as function of time during polymerization of VAc for synthesis VAc_1.5 (left) and synthesis VAc_1.8 (right).

**Figure 6 polymers-13-00669-f006:**
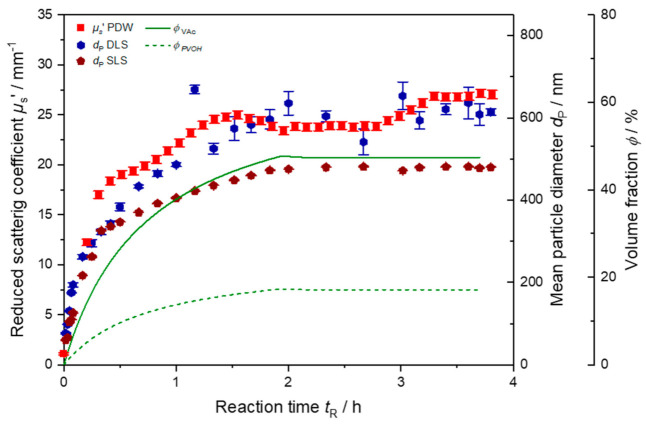
Inline *µ*_s_’ at *λ* = 637 nm and offline determined particle diameter from dynamic light scattering (DLS) (blue symbols) and static light scattering (SLS) (brown symbols). Volume fraction *ϕ*_VAc_ and *ϕ*_PVA_ (solid and dashed green line) in synthesis VAc_1.5.

**Figure 7 polymers-13-00669-f007:**
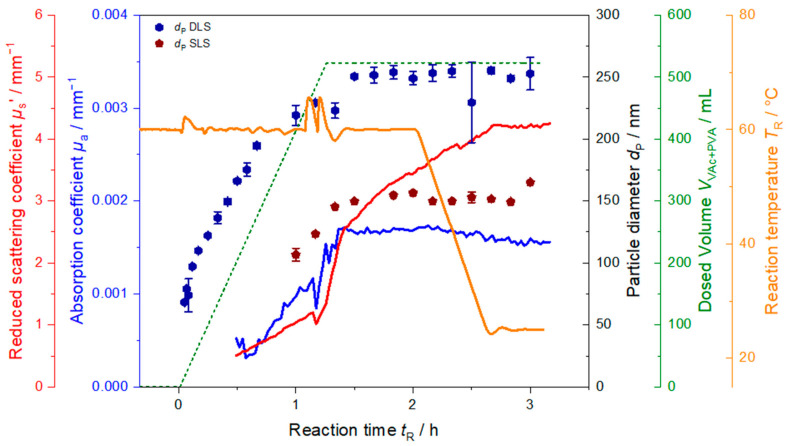
Reduced scattering coefficient *µ*_s_’ (red), absorption coefficient *µ*_a_ (blue) at λ = 637 nm, combined dosed mass of monomer and protective colloid (dashed green line) and reaction temperature (orange) as function of time during synthesis PVAc_1wr. The dosing of monomer and PVA had to be aborted due to gelation of the sample at *t*_R_ = 1.1 h.

**Table 1 polymers-13-00669-t001:** Total dosed volumes and feed rates of monomer, 16.7% (*w*/*w*) PVA-water solution and initiators listed for each synthesis.

Synthesis	Volume *V*_VAc_ /mL	Feed Rate VAc /mL min^−1^	Feed Rate PVA /mL min^−1^	Volume *V*_AA_ /mL	Volume *V*_NaPS_ /mL
VAc_1	350	4.52	2.46	31.66	31.66
VAc_1.5	525	4.52	1.64	44.55	44.55
VAc_1.6	560	4.52	1.53	50.66	50.66
VAc_1.8	630	4.52	1.36	53.33	53.33
VAc_2	700	4.52	1.23	63.33	63.33

**Table 2 polymers-13-00669-t002:** Final composition of the dispersion P Polyvinyl acetate (PVAc)_1wr with 36.3 g L ^−1^ PVA dissolved in 151.2 g of water in the initial charge after abortion of the dosing of monomer and PVA feeds.

Component	Dosed Volume V_1_ /mL	Target Volume V_2_ /mL	Fraction /%
Monomer VAc	339.16	350	96.90
PVA solution 15.7% (*w*/*w*)	184.125	190	96.90
Initiator solution 4.5% (*w*/*w*) NaPS	34.6	31.6	109.29
Initiator solution 3.5% (*w*/*w*) AA	34.6	31.6	109.29
Catalyst FAS	0.009	0.009	100
Total dispersion mass	743.96	745.67	

**Table 3 polymers-13-00669-t003:** Total volumes of dosed monomer V_VAc_, extrapolated density *ρ*_P_ and refractive index *n* (at *λ* = 637 nm) of the particles for each synthesis.

Synthesis	*V*_VAc_/mL	*ρ*_P_/g cm^−3^	*n* (*λ* = 637 nm)
VAc_1	300	1.2015 ± 0.0004	1.4797 ± 0.0007
VAc_1.5	525	1.2026 ± 0.0001	1.4827 ± 0.0010
VAc_1.6	560	1.2055 ± 0.0001	1.4851 ± 0.0011
VAc_1.8	630	1.2023 ± 0.0003	1.4906 ± 0.0007
VAc_2	700	-*^1^	-*^1^

*^1^ No determination of extrapolated particle density and refractive index possible, due to the high solid content and viscosity of the dispersion.

## Data Availability

Not applicable.

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
