# Peer review of "Process Characterization of Polyvinyl Acetate Emulsions Applying Inline Photon Density Wave Spectroscopy at High Solid Contents"

_polymers, 2021, doi:10.3390/polym13040669_

Round 1

Reviewer 1 Report

Polymers

Title:  Process characterization of polyvinyl acetate emulsions applying inline Photon Density Wave spectroscopy at high solid contents

Article Type: Original Article

We are pleased to send you moderate comments. The studied was done by professionals who deeply understand the essence of the subject. The results and theme of this article is quite interesting. Overall, the presentation style of article is acceptable for journal. However, the authors should consider following comments for its acceptance for publication.

  1. The first sentence of the abstract should be revise and write according to the next sentence.
  2. What is w at line 35?
  3. Line 25, Emulsion polymerization….. polymers used should be removed from this place, it looks wired at this place.
  4. The reference [4],[5] should be written [4,5], please check throughout the manuscript. There are several errors of references.
  5. What is the problem statement? I did not understand. The arrangement of the introduction is not good, please revise.
  6. 1. Synthesis, this section heading should be clearer. And relevant reference should be cite because it not new method.
  7. The term should be defined first before using the abbreviations.
  8. Regarding the replications, authors confirmed that replications were carried out. However, these results are not shown in the manuscript. How many replicated were carried out by experiment? Results seem to be related to a unique experiment. Please, clarify whether the results of this document are from a single experiment or from an average resulting from replications. If replicated were carried out, the use of average data is required as well as the standard deviation in the results and figures shown throughout the manuscript. In case of showing only one replicate, explain why only one is shown and include the standard deviations.
  9.  The line 219-221 is very confusing, please rewrite this line.
  10. In whole result and discussion, I found there is no comparison with other reported works, please compared the results with others.
  11. Section 5. Outlook and further research should be change into Conclusion and further research recommendations.
  12. The present section 5 is failed to deliver the concluding remarks please explain bit and add some future recommendations for more advance knowledge.
  13. There are several affirmations needs more reference support. Some are article are useful to cite such as Environmental application of smart polymer composites’’, ‘’Biological application of smart polymer composites’’ and ‘’ Preparation and characterization of nanosized lignin from oil palm (Elaeis guineensis) biomass as a novel emulsifying agent’’.

Overall, I am suggesting the major revision. The present research can attract the readers. Some above-mentioned corrections are needed to make the article more attractive and citable.

Reviewer 2 Report

The manuscript entitled “process characterization of polyvinyl acetate emulsions applying inline Photon Density Wave spectroscopy at high solid contents” by Schlappa et al. investigates the application of a relatively new technique concerning the inline study of a latex with a high solid amount with important practical application in industry.

The paper is clear, well written, and the conclusions are supported by the results. At the first view, this technique seems very interesting but I think that it must be improved before its use in industry.

  1. As a first remark, if the hydrolysis degree (DH) is smaller than 98%, the samples are generally abbreviated as poly(vinyl alcohol-co-vinyl acetate) or PVA. Poly(vinyl alcohol) or PVOH is used only for samples with DH > 98%.
  2. In the introduction section the authors must better discuss the role of the PVA, especially as a function of his DH. Some publications related to this theme must be cited: https://doi.org/10.1039/C4SM02766C ; https://doi.org/10.3390/polym3031065
  3. The DH of the PVA sample should be indicated in the materials section, even if it’s obvious from the name of the sample.
  4. Line 262: which values of VAc_1.8 were used for VAc_2?! I suppose that the authors refer to density and refractive index, no?!
  5. It will be of interest to have an idea about the conversion % and molar mass of samples as a function of time for all the experiments.
  6. Also, a photo of the experimental set-up might be useful for the readers, maybe in SI.
  7. Concerning the experimental set-up, it will be of interest to know at which point of the reactor, the samples are measured?! Which is the depth of the optical path?
  8. The authors have taken into account the hydrodynamic regime?!
  9. The non-invasive back scattering (NIBS) technique from Malvern allows the size determination of samples with a solid content up to 40%. Have the authors tried to analyze their samples without a dilution step?!

     In view of the above, I recommend the publication of the manuscript after major revisions.

Round 2

Reviewer 1 Report

In the revised manuscript authors carefully considered all of the raised comments and the manuscript is much improved. It can be accepted in the present form.

Reviewer 2 Report

The hydrodynamic regime concerns the agitation rate as it is know that the sizes of both the emulsion droplets and particles are influenced by the agitation regime. The paper can be published as it is.